# Integrated Bioinformatics and Machine Learning for Ascertainment and Validation of Biomarkers for Screening Breast Disease

**DOI:** 10.3390/genes16111389

**Published:** 2025-11-18

**Authors:** Qi Wang, Saisai Yang, Yao Zhang, Chengyu Piao, Xin Liu, Xiuhong Wu

**Affiliations:** 1National Key Laboratory, Heilongjiang University of Chinese Medicine, Harbin 150000, China; 18103696680@163.com (Q.W.); y1594797983@126.com (S.Y.); zyao991105@126.com (Y.Z.); anping0451@126.com (C.P.); 2State Key Laboratory of Multimodal Artificial Intelligence, Institute of Automation, Chinese Academy of Sciences, Beijing 100000, China; xinl2021@163.com

**Keywords:** benign breast disease, breast cancer, bioinformatics, machine learning, immune cell infiltration

## Abstract

**Background:** This research sought to screen potential biomarkers in diagnosing breast diseases and elucidating their immune-related mechanisms. **Methods:** Three datasets were attained from the Gene Expression Omnibus (GEO) database. LIMMA package and weighted gene co-expression network analysis (WGCNA) were used to ascertain differentially expressed genes (DEGs) and key modules in benign breast disease (BBD) and breast cancer (BC). The intersecting genes underwent functional enrichment analysis. Three machine learning (ML) methods (encompassing LASSO regression, random forest, and support vector machine recursive feature elimination (SVM-RFE)) were implemented to select core genes. The diagnostic performance of the core genes was evaluated by comparing their expression levels, plotting receiver operating characteristic (ROC) curves, and constructing a Nomogram. The TCGA-BRCA dataset was used to estimate the prognostic capability of the core genes among individuals with BC. Finally, the IC infiltration was ascertained utilizing the CIBERSORT algorithm. **Results:** In total, 2579 DEGs were identified in BBD. WGCNA exhibited that the 1652 genes in green and pink modules were strongly correlated with BBD. In BC, 2742 DEGs were identified. The turquoise and red modules contained 7286 genes exhibiting strong correlations with BC. After intersecting, 41 common genes were obtained, which were predominantly enriched in immune and inflammation regulation pathways. Through integrated screening with three ML algorithms, Arrestin Domain Containing 1 (*ARRDC1*) and ATPase Sarcoplasmic/Endoplasmic Reticulum Ca^2+^ Transporting 2 (*ATP2A2*) were identified as core genes. The ROC curve exhibited that the AUC for the two genes was greater than 0.8. The calibration curve of the nomogram signified a strong alignment between the anticipated risk and detected results. Survival analysis in TCGA-BRCA showed that the high expression of the two genes exhibited a significantly positive association with unfavorable prognosis. Immune infiltration analysis further demonstrated the dysregulation of multiple immune cells in patient samples. **Conclusions:**
*ARRDC1* and *ATP2A2* are strongly linked to BBD and BC. These findings might enhance our comprehension of the pathogenesis and progression of both BBD and BC, offering prospective biological biomarkers and therapeutic targets for clinical treatment.

## 1. Introduction

Breast cancer (BC) represents the prevalent malignant tumor in females globally. Based upon cancer statistics for 2023, BC accounts for 23% of all cancers [1]. In 2020, approximately 2.26 million females were diagnosed with BC, representing 15.5% of cancer-associated mortalities among females. In developed countries, about one in eight women might develop BC [2].

Benign breast disease (BBD) encompasses a variety of distinct breast lesions [3]. These conditions draw clinical attention due to abnormal findings on imaging or physical examinations. Clinically, BBD is more common than malignant breast diseases. Benign epithelial lesions of the breast are histologically sorted into three categories: non-proliferative, non-atypical proliferative, and atypical proliferative lesions. Epidemiological findings suggest that non-proliferative lesions are generally not correlated with elevated risks of BC, while atypical proliferative lesions might elevate risks of developing BC [4]. Dupont et al. [5] conducted a case–control study on participants of the BC Detection Demonstration Project (BCDDP), evaluating the impact of proliferative breast disease (PD) on BC risk. Biopsy results exhibited that females with atypical hyperplasia had a 4.3-fold elevated risk of BC relative to women with non-PD. Although the etiology of benign breast lesions remains unclear, factors like a family history of BC, hormonal influences, and age all affect the risk of developing BBD [6].

Currently, histopathological examination (including core needle biopsy, fine needle aspiration cytology, and surgical excision biopsy) remains the gold standard for diagnosing BBD and malignant breast diseases [7]. A study from the Norwegian Breast Screening Program indicated that among women undergoing biopsy, approximately 50% were diagnosed with benign lesions [8]. However, these methods are invasive and rely on invasive procedures. Therefore, discovering highly sensitive and specific peripheral blood biomarkers holds critical clinical and translational value, enabling early screening, differential diagnosis, and fewer unnecessary invasive procedures in breast diseases (Appendix A).

This research integrated transcriptomic data for BBD and BC from the Gene Expression Omnibus (GEO) database. Differential expression analysis and weighted gene co-expression network analysis (WGCNA) were implemented to ascertain differentially expressed genes (DEGs) and key modules. Three machine learning (ML) methods were subsequently applied to discern core genes shared between the two diseases. The findings might provide guidance for the prediction of breast diseases and the mechanisms of disease progression, offering new therapeutic targets for treating breast diseases.

## 2. Methods

### 2.1. Microarray Data

Three publicly attainable microarray datasets (GSE42568 [9], GSE27562 [10], and GSE61304 [11,12]) were retrieved from the National Center for Biotechnology Information (NCBI) GEO database (https://www.ncbi.nlm.nih.gov/, accessed on 13 June 2025). GSE42568, GSE27562, and GSE61304 were all based upon the GPL570 platform. The GSE27562 dataset encompassed 37 benign samples and 31 normal samples. The GSE42568 dataset encompassed 104 BC samples and 17 normal samples. The GSE61304 dataset encompassed 58 breast tumor samples and 4 normal samples. The Combat function from the SVA package (version 3.56.0) was leveraged to correct batch effects (BEs) in the gene expression data of GSE42568 and GSE61304. The effectiveness of BE removal was estimated utilizing distribution boxplots and principal component analysis (PCA).

The TCGA-BRCA dataset was attained from the TCGA database (https://portal.gdc.cancer.gov/) [13], which encompassed 1106 BC samples, 113 adjacent tissue samples, and corresponding clinical information. Detailed information about the dataset is provided in Table 1. Appendix A lists the abbreviations used in the text along with their corresponding full forms.

### 2.2. Differential Expression and Enrichment Analysis

The limma R package (version 3.64.1) was used to ascertain DEGs between disease and normal cohorts. For the GSE27562 dataset, the DEG threshold was designated as adj.*p* < 0.05 and |fold change| > 1.2. For the integrated data of GSE42568 and GSE61304, the DEG threshold was set to adj.*p* < 0.05 and |fold change| > 2. Subsequently, volcano plots were used to present the differential analysis results for every cohort. In the plots, blue signified low expression, and red signified high expression. To validate the biological functions and related signaling pathways of key genes, we intersected the DEGs from the BBD and BC groups to attain intersecting genes. Using the R packages clusterProfiler (version 4.10.1), org.Hs.eg.db (version 3.21.0), and enrichplot (version 1.22.0), GO [14] and KEGG [15] enrichment analyses were performed on the obtained intersection genes.

### 2.3. WGCNA

WGCNA was a systems biology method utilized to elucidate gene correlation patterns within microarray data [16]. For each disease, the WGCNA R (version 1.73) package was leveraged following the steps below: (i) a sample clustering tree was plotted to check for outlier samples; (ii) network topology properties were computed at different soft threshold values and the optimal power was selected; (iii) the Topological Overlap Matrix (TOM) similarity was calculated and converted to a distance matrix; (iv) dynamic pruning was performed based on the gene dendrogram (geneTree) and dissTOM to identify gene modules; (v) correlations between module eigengenes were calculated, hierarchical clustering was performed, and the clustering dendrogram of module eigengenes was visualized; (vi) the interrelation between modules and traits was ascertained and the results were visualized. Finally, the disease-associated module genes were attained.

### 2.4. Core Gene Screening

Three ML algorithms (LASSO: least absolute shrinkage and selection operator; SVM-RFE: support vector machine-recursive feature elimination; RF: random forest) were used to further screen the shared core genes between the two diseases. After screening with the aforementioned three algorithms, core genes were ascertained by taking the intersection of results from each algorithm. The pROC package (version 1.18.5) was used to construct the receiver operating characteristic (ROC) curve to estimate the diagnostic accuracy of the identified genes in the discovery cohort. Kaplan–Meier curves were plotted utilizing the survival package in R (version 3.5.8) to exhibit the survival differences between high and low expression groups of core genes in the TCGA-BRCA cohort. In the training set, a nomogram was built based upon logistic regression analysis. The forecasting capability of the nomogram was estimated utilizing ROC curves and calibration curves. Furthermore, after attaining the diagnostic genes, gene set enrichment analysis (GSEA) for every diagnostic gene in both groups was implemented utilizing the ClusterProfiler (version 4.10.1) and enrichplot R (version 1.22.0) packages.

### 2.5. Immune Infiltration Analysis

CIBERSORT analysis was implemented for each disease sample to ascertain the relative levels of immune cells (ICs). The CIBERSORT algorithm parsed IC composition through deconvolution based upon gene expression data [17]. Based upon the information from the CIBERSORT website (http://cibersort.stanford.edu/), LM22 contained 22 annotated gene signatures. Utilizing CIBERSORT with 1000 iterations, quantitative analysis of 22 IC categories was implemented based upon the LM22 gene signature. For each sample, the output estimates generated by CIBERSORT were normalized to sum to 1, facilitating comparisons across different IC types and datasets. Moreover, correlation analysis was implemented between infiltrating ICs and diagnostic targeting biomarkers.

### 2.6. Statistical Analysis

All analyses were implemented leveraging R (version 4.3.3). Non-parametric tests were implemented to contrast continuous variables between two cohorts. The Spearman’s rank correlation test was used to ascertain the interrelation between gene expression and IC scores. Statistical significance was set to *p* < 0.05. The study design flowchart is presented in Figure 1.

## 3. Results

### 3.1. Identification of DEGs

Prior to analysis, BE in the gene expression data were corrected utilizing the ComBat algorithm. Data distributions before and after correction were visualized with boxplots (Figure 2A,B) and PCA (Figure 2C,D). DEG analysis between cohorts was implemented utilizing the limma R package. Within the BBD group, 2579 DEGs were identified (1168 downregulated, 1411 upregulated; Figure 3A), while the BC group exhibited 2742 DEGs (1261 upregulated, 1481 downregulated; Figure 3B). Intersecting upregulated and downregulated genes between the two groups were subsequently visualized utilizing Venn diagrams (Figure 3C,D).

### 3.2. Enrichment Analysis of DEGs

Functional enrichment analysis was implemented on DEGs to validate the functions of potential targets.

GO enrichment analysis denoted that the DEGs were primarily enriched in the subsequent categories: (i) Biological process (BP): nuclear chromosome segregation, chromosome segregation, sister chromatid segregation, mitotic nuclear division, and nuclear division; (ii) Cellular component (CC): spindle, nuclear periphery, nuclear matrix, chromosomal region, and ubiquitin ligase complex (Figure 4A).

KEGG pathway enrichment results exhibited that these genes were predominantly enriched in the cell cycle, oocyte meiosis, p53 signaling pathway, DNA replication, and progesterone-mediated oocyte maturation (Figure 4B). These findings denoted that the dysregulation of cell division and chromosome segregation processes was a common feature in breast disease states. Notably, these pathways had profound and close associations with immune and inflammatory responses, serving as key drivers in the activation of the immune microenvironment.

### 3.3. WGCNA Screening of Key Modules

To determine associations between disease states and key genes beyond differential expression analysis, we implemented WGCNA. In the BBD group, based on scale independence and average connectivity, a fitting index >0.85 was deemed to be a scale-free topology, and β was designated to 13 (Figure 5A). Utilizing the adjacency function, an adjacency matrix was produced. Hierarchical clustering was implemented utilizing dissTOM metrics (Figure 5B). In total, 11 modules were ascertained and labeled with unique colors. We analyzed the correlation between the feature genes and the phenotype (BBD or control samples) and detected that two modules had a strong correlation with BBD: the green (cor = 0.49, *p* = 2 × 10^−5^) and pink (cor = 0.41, *p* = 5 × 10^−4^) modules (Figure 5C).

Scatter plots depicted the interrelation between gene significance (GS) and module membership (MM) within the green and pink modules (Figure 5D,E). In the green and pink modules, a significant correlation was observed between GS and MM, with correlation coefficients of 0.27, *p* = 1.9 × 10^−21^ for the green module and 0.22, *p* = 2.2 × 10^−6^ for the pink module (Figure 5D,E). Furthermore, we applied WGCNA to the BC group with β set at 4 (Figure 5F). In total, 18 modules were ascertained, with the red and turquoise modules exhibiting strong interrelations (Figure 5G,H). Additionally, we ascertained the interrelation between GS and MM in the red and turquoise modules (Figure 5I,J). Subsequently, by intersecting the DEGs with the genes identified by WGCNA, 41 key disease-associated marker genes were identified (Figure 5K).

### 3.4. Core Gene Screening Based upon ML

To ascertain hub genes correlated with BBD and BC, we applied three ML algorithms to genes overlapping between WGCNA modules and DEGs. In the BBD group, 10-fold cross-validation (CV) was employed to modify the parameters and mitigate overfitting. The LASSO regression analysis revealed that the λ value corresponding to the minimum CV error was 0.01488008. The model with 17 genes was the optimal predictor (Figure 6A). The RF algorithm was implemented to further identify diagnostic candidates that could optimize classifier performance. Figure 6B illustrates the evolution of the random forest error rate (with 95% confidence intervals) across increasing numbers of decision trees in the training cohort. Figure 6C displays the importance ranking of individual genes. We selected MeanDecreaseGini > 1 as the threshold, and 12 genes were identified. Moreover, the SVM-RFE ascertained 24 genes with the lowest 10-fold CV error and optimal 10-fold CV accuracy (Figure 6D). Finally, by overlapping the results from the three methods, four consensus biomarkers for the abnormal breast group were identified (Figure 6E).

Similarly, by setting λ to 0.004105665 in the LASSO algorithm, 11 feature genes were obtained in the BC group (Figure 6F). Figure 6G,H exhibit the results of the RF. Fourteen genes were ascertained utilizing the threshold of MeanDecreaseGini > 1. Subsequently, a subset of 19 core genes was identified using the SVM-RFE algorithm (Figure 6I). The overlap among the five biomarkers is depicted in Figure 6J.

Arrestin Domain Containing 1 (*ARRDC1*) and ATPase Sarcoplasmic/Endoplasmic Reticulum Ca^2+^ Transporting 2 (*ATP2A2*) were shared by all three algorithms and were therefore designated as the final core genes correlated with BBD and BC (Figure 6K).

### 3.5. Validation of Core Genes

Relative to normal controls, *ARRDC1* and *ATP2A2* were significantly upregulated in individuals with BBD or BC (Figure 7A,B). Moreover, we ascertained the area under the ROC curve (AUC-ROC) for every core gene. In the BBD group, *ARRDC1* had an AUC of 0.813, and *ATP2A2* of 0.843 (Figure 7C). The same ROC analysis was implemented for the BC group, where *ARRDC1* (AUC = 0.936) and *ATP2A2* (AUC = 0.971) demonstrated high diagnostic efficiency (Figure 7D). These AUC-ROC values denoted that the core genes had high diagnostic accuracy for predicting BBD and BC.

To further ascertain interrelations between *ARRDC1* and *ATP2A2* with prognosis among individuals with BC, we conducted survival analysis for high and low expression cohorts of these genes in the TCGA dataset. Marked distinctions were detected in survival probabilities between the high and low expression cohorts of *ARRDC1* and *ATP2A2* (Figure 8). High expressions of *ARRDC1* and *ATP2A2* were linked to a trend of unfavorable clinical prognosis. This denotes that high expressions of *ARRDC1* and *ATP2A2* were positively correlated with poorer survival probabilities. These findings further support the diagnostic value of the ascertained core genes.

### 3.6. Construction of the Nomogram

To further validate the forecasting capability of *ARRDC1* and *ATP2A2* for individuals with BBD or BC, a nomogram was constructed based on the two final-selected genes (*ARRDC1* and *ATP2A2*) (Figure 9A,D). The calibration curves were developed utilizing 1000 bootstrap resamples from two cohorts, demonstrating high consistency between the anticipated risks by the nomogram and the observed outcomes (Figure 9B,E). ROC curves were used to estimate the forecasting precision of the nomogram. In the BBD cohort, the AUC was 0.880. In the BC cohort, the AUC was 0.988, denoting that *ARRDC1* and *ATP2A2* performed well in diagnosing BBD and malignant breast diseases (Figure 9C,F).

### 3.7. GSEA Analysis

This research implemented single-gene GSEA analysis for these two biomarkers in BBD and BC datasets. Figure 10 shows that *ARRDC1* is significantly correlated with hormone signaling, apoptosis, the notch signaling pathway, and microRNAs in cancer pathways in both disease groups. *ATP2A2* is significantly correlated with the apoptosis, p53 signaling pathway, and hormone signaling pathways (Figure 11). These findings shed light on the underlying mechanisms of BBD and BC, with particular emphasis on their co-enrichment in the apoptosis and hormone signaling pathways. This provides key clues for understanding their functions in BBD and malignant breast lesions.

### 3.8. Immunoanalysis

Immunological function was crucial in the occurrence and progression of BBD and BC. To ascertain the effects of *ARRDC1* and its co-expressed gene *ATP2A2* in the immune regulation of BC and BBD, we analyzed the infiltration differences of 22 IC subtypes between the disease groups and normal control tissues utilizing the CIBERSORT algorithm.

In the BBD group, CIBERSORT analysis revealed marked distinctions in the immune microenvironment between BBD tissues and normal breast tissues (Figure 12A). Specifically, the infiltration levels of NK.cells.activated (*p* < 0.01) and plasma.cells (*p* < 0.0001) were considerably elevated in BBD tissues. The infiltration levels of neutrophils (*p* < 0.01) and T.cells.CD4.naive (*p* < 0.01) were considerably diminished. These results denote that the most prominent feature of the BBD microenvironment was the significant enrichment of NK.cells.activated and plasma.cells.

We estimated the interrelation between the expression of *ARRDC1* and *ATP2A2* and the changes in IC subpopulations through correlation analysis. The expression of *ARRDC1* was positively linked to the infiltration of NK.cells.activated and negatively linked to the infiltration of T.cells.CD4.naive (Figure 12C). The expression of *ATP2A2* exhibited a highly consistent pattern (Figure 12D), showing a positive association with the infiltration of NK.cells.activated and a negative association with the infiltration of T.cells.CD4.naive. Natural killer (NK) cells serve as the primary effector cells against cancer in innate immunity and exhibit high heterogeneity within the tumor microenvironment [18]. *ARRDC1* and its co-expressed gene *ATP2A2* were both associated with the infiltration of activated NK cells. These results denote that *ARRDC1* and its co-expressed gene *ATP2A2* might be correlated with a specific immune status manifested as the activation of innate immunity in BBD [19].

In the BC group, the violin plots exhibited considerably different IC types relative to the normal group (Figure 13A). The infiltration levels of Macrophages.M0 (*p* < 0.0001), Macrophages.M1 (*p* < 0.0001), and T.cells.CD8 (*p* < 0.01) were considerably elevated, while the infiltration levels of Macrophages.M2 (*p* < 0.0001), Mast.cells.activated (*p* < 0.0001), Monocytes (*p* < 0.01), and NK.cells.activated (*p* < 0.0001) were considerably diminished (Figure 13B). The expression levels of *ARRDC1* and *ATP2A2* demonstrated highly consistent immunoregulatory patterns (Figure 13C,D). Both genes demonstrated significant positive correlations with the infiltration levels of Macrophages.M0, T.cells.follicular.helper, and NK.cells.resting. Meanwhile, they showed significant negative correlations with the infiltration levels of T.cells.CD4.memory.resting, Mast.cells.resting, Monocytes, and NK.cells.activated. These results denote that *ARRDC1* and its co-expressed gene *ATP2A2* might collaboratively be involved in the formation and regulation of the immune-suppressive microenvironment in BC.

## 4. Discussion

Breast symptoms, including lumps, nipple discharge, and pain, attract attention and prompt more than 15 million people to seek medical care annually. Although 90% of these symptoms are caused by BBD, BC remains a constant concern. Meanwhile, mammography screening for early breast cancer detection continues to face challenges. Consequently, people with breasts are more likely to have symptoms and/or more advanced disease [20]. Thus, breast health remains critical for women, underscoring the importance of early screening.

In this study, the DEGs obtained from the screening dataset were subjected to KEGG and GO enrichment analyses. The GO enrichment analysis denoted that in the disease groups, the DEGs were considerably enriched in BP related to cell division, chromosome segregation, and cell cycle regulation. Chromosomal mis-segregation induces further genomic instability, resulting in cell cycle arrest [21]. Persistent errors in chromosome segregation during mitosis cause chromosomal instability (CIN) [22]. BC is inherently a disease of genomic instability. CIN is its primary manifestation and is strongly linked to tumor evolution, invasion, and poor prognosis [23]. KEGG enrichment analysis denotes that the DEGs were correlated with several signaling pathways and were considerably enriched in the cell cycle and p53 signaling pathways. In BC, the p53 pathway is typically inactivated due to mutations or functional suppression. This implies that genetically unstable cells generated by CIN are not effectively cleared, enabling their survival and the accumulation of additional mutations. This ultimately triggers the formation and development of malignant tumors [24].

By integrating bioinformatics and ML algorithms, we identified two key genes (*ARRDC1* and *ATP2A2*) and demonstrated their diagnostic performance through ROC analysis. Significant distinctions in the abnormal expression of the two genes between the disease and normal cohorts suggested that they are crucial in the development of breast diseases. The full name of *ARRDC1* is Arrestin Domain Containing 1, which encodes a scaffold protein containing an arrestin domain [25]. As a member of the α-arrestin family, this protein functions primarily as a molecular adaptor that mediates ubiquitination and the degradation of membrane proteins. It also participates in a specialized form of micropinocytosis and unique extracellular vesicle formation [26,27]. Researchers initially discovered mechanisms related to intracellular retention and protein sorting in model organisms such as yeast. In mammals, *ARRDC1* has been identified as a crucial component of this mechanism, serving as an important molecule that regulates the stability of cell membrane proteins and signal transduction [28]. Its role in various physiological and pathological processes has garnered increasing attention. Studies have found that *ARRDC1* plays a pivotal role in the malignant biological behaviors of cancer. Also, the abnormal expression or dysfunction of *ARRDC1* is closely associated with the onset and progression of various diseases, including diabetes, colorectal cancer, and liver cancer [28,29,30]. Li et al. [31] measured the expression of *ARRDC1-AS1*, *miR-4731-5p*, and *AKT1* in BC cell lines and discovered that extracellular vesicles containing *ARRDC1-AS1* (BCSCs-EVs) promote the malignant characteristics of BC cells. Liu et al. [32] found that upregulation of *ARRDC1-AS1* could predict the recurrence of BC. These findings are consistent with our results, indicating that the overexpression of *ARRDC1* reflects a more severe state of disease. Therefore, *ARRDC1* may serve as a reliable biomarker for breast diseases.

As a member of the *ATP2As* gene family, *ATP2A2* encodes one type of sarcoplasmic (endoplasmic) reticulum calcium transport ATPase, namely SERCA2b [33]. Mutations in the human *ATP2A2* gene, which encodes housekeeping isoforms of the endoplasmic reticulum (SERCA2) and secretory pathway Ca^2+^-ATPase (SPCA1) pumps, cause autosomal dominant genetic skin disorders. Darier’s disease (DD) is caused by heterozygous variants in the *ATP2A2* gene [34]. Studies have shown that abnormal expression of *ATP2A2* disrupts intracellular calcium ion homeostasis, leading to disturbances in calcium ion-related signaling pathways. This promotes malignant cell proliferation, enhances tumor migratory capacity, and stimulates angiogenesis, ultimately contributing to the initiation and progression of tumors [35]. Reports have confirmed that abnormal expression of *ATP2A2* is involved in the occurrence of lung cancer [34], the differentiation of squamous cancer cells [34], and early events in the development of colon cancer [36]. However, functional studies on the *ATP2A2* gene in breast diseases remain relatively scarce. Wang et al. [37] identified several proteins, including SERPINE1, RPL26L1, PLOD2, UGDH, LGALS1, VIM, TSTA3, SORD, FDFT1, and *ATP2A2*, which are involved in epithelial–mesenchymal transition (EMT) programs and may also participate in normal human breast development. Partial activation of EMT drives metastasis and the dissemination of breast tumors to distant organs. The findings of this study indicate that the expression of *ATP2A2* is upregulated in patients with breast diseases (especially BC), and the overexpression of *ATP2A2* is associated with a poorer survival rate. Consequently, the *ATP2A2* gene exhibits potential as a biomarker and therapeutic target for breast diseases.

The results of GSEA revealed that *ARRDC1* and *ATP2A2* are co-enriched in multiple signaling pathways closely associated with the pathogenesis of breast diseases in both BBD and BC. These pathways include hormone signaling, apoptosis, the Notch signaling pathway, microRNAs in cancer, the p53 signaling pathway, and others. *ARRDC1* primarily regulates hormone receptor signal transduction and Notch pathway activity, influencing the differentiation of breast epithelial cells. The Notch pathway is a key signaling pathway that maintains the self-renewal and multipotent differentiation capability of BC stem cells. Its abnormal activation directly promotes the recurrence and metastasis of tumors [38]. Studies have shown that *ARRDC1* may influence the activity of this pathway by regulating the endocytosis and lysosomal degradation of Notch receptors, thereby participating in the transformation process of breast epithelial cells [39]. Meanwhile, *ATP2A2* maintains calcium homeostasis, thereby influencing p53-dependent apoptotic signaling and hormonal responses to ensure the homeostasis of breast tissues [33]. As a classic tumor-suppressor gene, the p53 gene can disrupt the expression of miRNA when mutated or degraded, thereby increasing the risk of BC [40]. The coordinated dysregulation of these pathways disrupts the normal proliferation–apoptosis balance, ultimately leading to the development of breast diseases. This highlights the central roles of *ARRDC1* and *ATP2A2* in the pathogenesis of breast.

In addition, the hormone signaling pathways jointly participated in by *ARRDC1* and *ATP2A2* are crucial in the development of breast diseases. Breast is a hormone-responsive organ. The development and function of the breast are regulated by multiple hormonal signals. Both genes are enriched in this pathway, suggesting that they cooperatively maintain hormonal homeostasis through distinct mechanisms. *ARRDC1* modulates membrane-receptor signaling, whereas *ATP2A2* acts through a calcium signal-mediated second messenger system [41]. The imbalance of this multi-layered regulatory network may represent a specific pathogenic mechanism underlying breast diseases.

Subsequently, immune infiltration analysis was conducted utilizing the CIBERSORT method. The results revealed distinctions in the relative abundance of various IC types between BBD and BC. Also, interrelations between core genes and ICs were detected. The *ARRDC1* and *ATP2A2* genes are primarily associated with macrophages, monocytes, follicular helper T cells, and NK cells, thereby reinforcing the link between immune infiltration and the tumor microenvironment. Therefore, if aberrant expression of these two genes is detected in blood samples during routine check-ups, clinicians should be highly alert to the possibility of breast disease and pay particular attention to the risk of developing BC.

This research has several limitations. First, as the data utilized in this research were derived from public databases, we lacked suitable data for verifying BBD. Consequently, larger, multi-center cohorts are required for subsequent validation to ensure the generalizability of the findings. Furthermore, we applied an integrated bioinformatics method to screen for core genes and propose their potential functions, but direct evidence supporting a regulatory or causal relationship between key genes and diseases remains lacking. Therefore, subsequent studies should involve experimental validation of the core genes and in-depth mechanistic investigation to confirm their functional roles in the disease.

## 5. Conclusions

In conclusion, this study implemented bioinformatics and ML algorithms to ascertain two key biomarkers (*ARRDC1* and *ATP2A2*) for screening breast diseases, including BBD and BC. This research might provide novel potential strategies for the early screening and clinical diagnosis of breast diseases. The identified key molecules represent promising therapeutic targets with clinical translational potential. Moreover, the findings offer insights into the molecular mechanisms underlying this disease.

## Figures and Tables

**Figure 1 genes-16-01389-f001:**
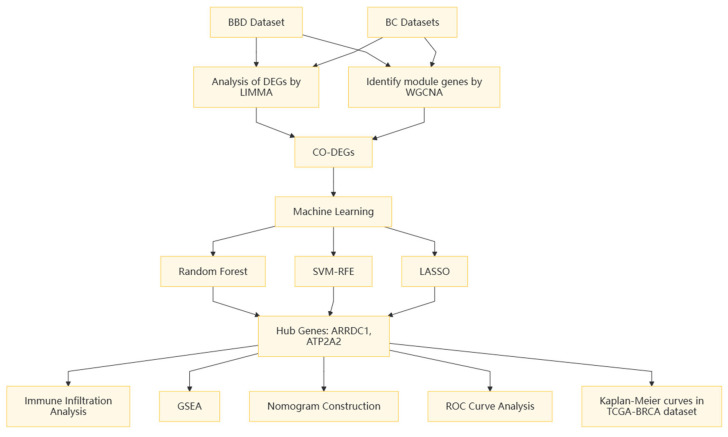
Research design flowchart.

**Figure 2 genes-16-01389-f002:**
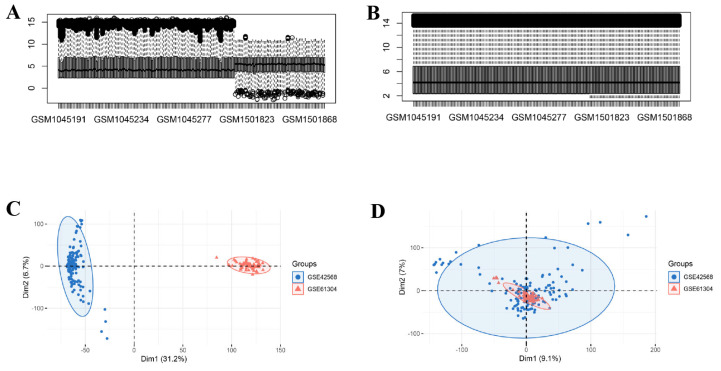
Batch effect removal for GSE42568 and GSE61304. (**A**) Distribution plot of the GEO dataset prior to batch correction; (**B**) Distribution plot of the GEO dataset upon batch correction, in the box plot, the black lines indicate the median, and the black circles mark the outliers; (**C**) PCA plot of the GEO dataset prior to batch correction; (**D**) PCA plot of the GEO dataset upon batch correction. Note: Red and blue grids denote upregulated and downregulated DEGs.

**Figure 3 genes-16-01389-f003:**
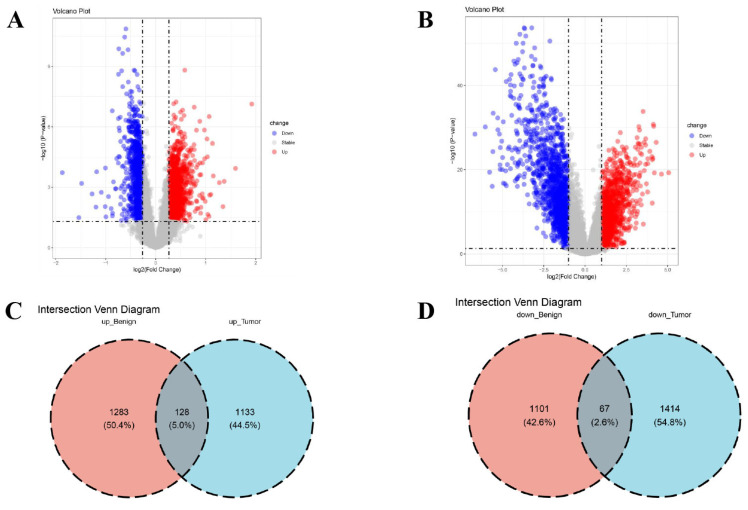
Ascertainment of DEGs in BBD and BC. (**A**) Volcano plot (VP) of DEGs in the BBD group; (**B**) VP of DEGs in the BC group, in the volcano plot, the dotted line on the vertical axis represents an adjusted *p*-value of 0.05, while the dotted line on the horizontal axis indicates that the absolute value of log_2_ (Fold Change) is equal to 1; (**C**) Venn diagram exhibiting the intersection of upregulated genes within the two datasets; (**D**) Venn diagram exhibiting the intersection of downregulated genes within the two datasets.

**Figure 4 genes-16-01389-f004:**
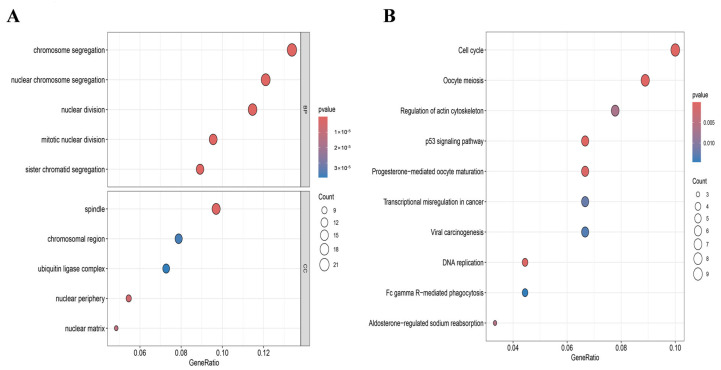
Enrichment analysis of DEGs. (**A**) GO analysis of DEGs; (**B**) KEGG pathway analysis of DEGs.

**Figure 5 genes-16-01389-f005:**
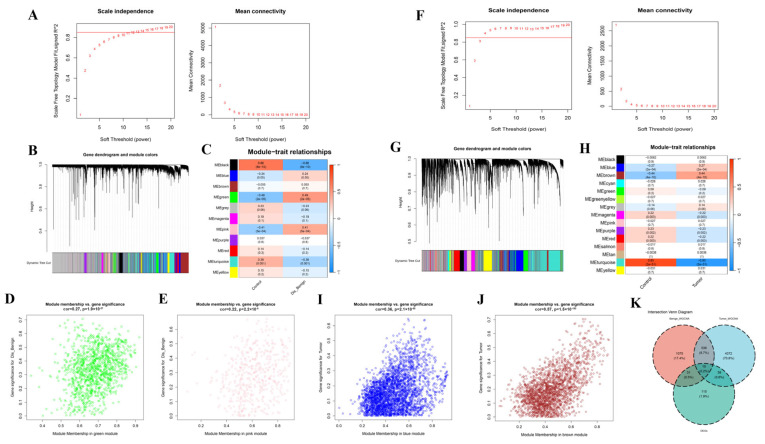
Screening of Disease Marker Genes in BBD and BC Based upon WGCNA. (**A**) Ascertainment of the soft threshold index for BBD, the horizontal red line represents the empirical threshold (=0.85) for the scale-free fitting index; (**B**) Clustering dendrogram of BBD genes with high connectivity in principal modules, in the line chart, the black solid line represents the fitted trend curve of module connectivity; the black dotted line represents the baseline of the scale-free fitting index; (**C**) Heatmap of interrelations between module and phenotype in BBD. This represents the interrelation between modules and the phenotype in BBD. Every cell contains the interrelation coefficient and *p*-value; (**D**) Gene significance scatter plot for the green module. The interrelation between module membership and gene significance: analysis of gene significance and disease association in the green module; (**E**) Gene significance plot for the pink module. The interrelation between genes and disease characteristics in the pink module: exhibiting the interrelation between gene expression and disease traits in the pink module; (**F**) Calculation of the soft threshold power for BC; (**G**) Clustering dendrogram of BC modules with highly connected genes; (**H**) Interrelations between module and trait in BC. Every cell encompasses the interrelation and *p*-value; (**I**) Gene significance plot for the red module; (**J**) Gene significance plot for the turquoise module; (**K**) Venn diagram of intersecting genes between WGCNA and DEGs.

**Figure 6 genes-16-01389-f006:**
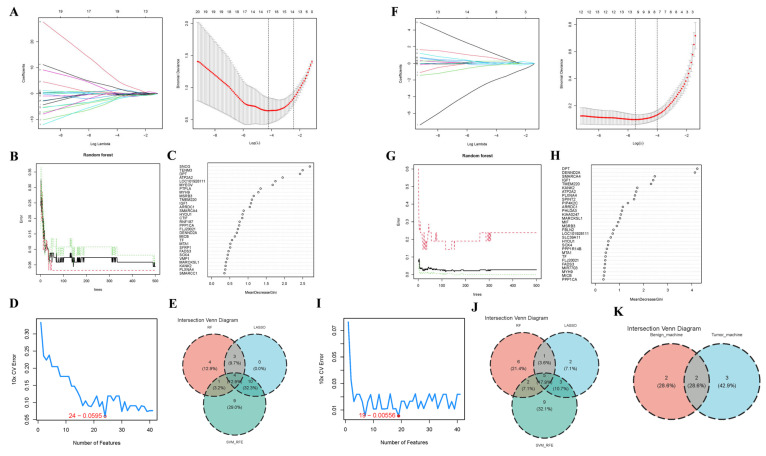
Core Gene Screening for BC and BBD Utilizing Three Machine Learning (ML) Algorithms. (**A**) In the BBD dataset, LASSO regression analysis was applied to the 41 intersecting genes to select biomarkers. The coefficient plot of the LASSO model exhibits the final parameter ascertainment of λ (lambda), the left dotted line represents log10(lambda.min), and the right dotted line represents log10(lambda.1se); (**B**) The effect of the quantity of decision trees on the diagnostic error rate in the RF algorithm, the black solid line represents the Overall OOB error, the red dotted line represents the Overall OOB error for the control group, and the green dotted line represents the Overall OOB error for the Benign group; (**C**) A scatter plot displaying the importance scores of the 41 feature genes in the RF model; (**D**) The line chart exhibits that the lowest SVM cross-validation error (0.0595) is achieved when the feature number is 24; (**E**) Intersection of three ML algorithms utilizing Venn diagrams yielded seven DEGs validated as candidate BBD biomarkers; (**F**) In the BC dataset, the coefficient plot of the LASSO model exhibited the final parameter ascertainment of λ (lambda); (**G**) The interrelation between error rate and the number of trees in the RF algorithm; (**H**) Ranking of genes based upon importance score; (**I**) In BC, 19 core genes were selected utilizing the SVM-RFE algorithm; (**J**) The intersection of the three ML algorithms was ascertained utilizing the Venn tool; (**K**) Venn diagram for the feature selection in BBD and BC based upon ML.

**Figure 7 genes-16-01389-f007:**
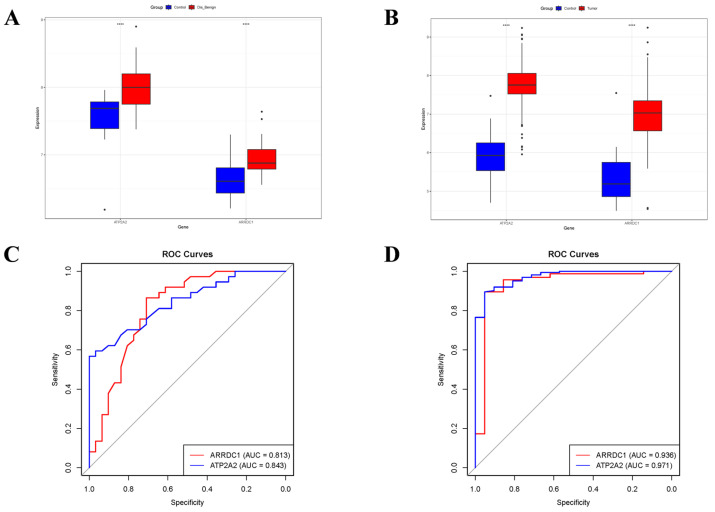
Diagnostic value of candidate core biomarkers evaluated by expression profiles and ROC Curve. (**A**) Differential expression of two genes (*ARRDC1* and *ATP2A2*) in the BBD dataset (GSE27562); (**B**) Differential expression of two genes (*ARRDC1* and *ATP2A2*) in the integrated BC dataset, in the box plot, the black points represent the corresponding Youden’s indices at different cutoff points; (**C**) ROC curves of *ARRDC1* and *ATP2A2* in the BBD dataset (GSE27562); (**D**) ROC curves of *ARRDC1* and *ATP2A2* in the integrated BC dataset, the gray diagonal line represents the reference line for random guessing with no diagnostic value (AUC = 0.5). Note: * *p* < 0.05, ** *p* < 0.01, *** *p* < 0.001, **** *p* < 0.0001.

**Figure 8 genes-16-01389-f008:**
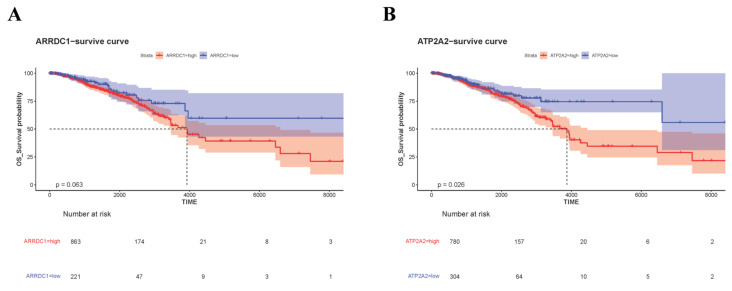
Kaplan–Meier survival curves in TCGA-BRCA. (**A**) Survival curve for *ARRDC1*; (**B**) Survival curve for *ATP2A2*. The dotted lines in the figure are used to highlight the median survival time.

**Figure 9 genes-16-01389-f009:**
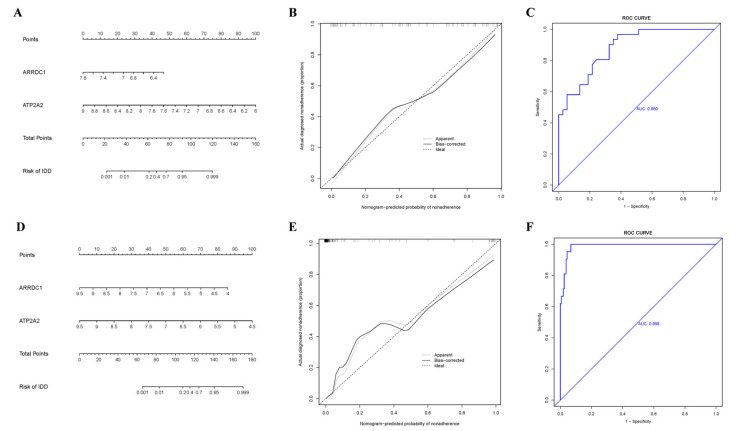
Nomogram and calibration curves. (**A**) Nomogram for forecasting the occurrence of BBD; (**B**) Calibration curve for BBD; (**C**) ROC curve for the BBD dataset; (**D**) Nomogram for forecasting the occurrence of BC; (**E**) Calibration curve for BC; (**F**) ROC curve for the BC dataset.

**Figure 10 genes-16-01389-f010:**
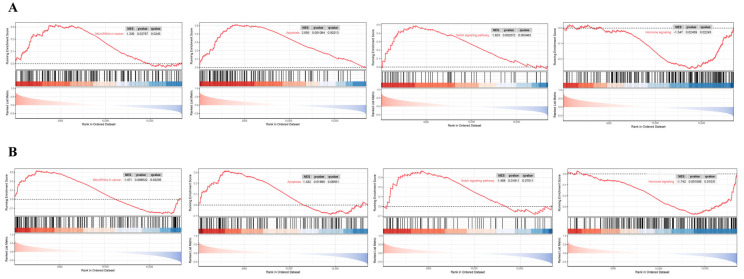
GSEA for the *ARRDC1* diagnostic gene. (**A**) GSEA analysis of the *ARRDC1* gene in the BBD group; (**B**) GSEA analysis of the *ARRDC1* gene in the BC group. The red line in the figure represents the running enrichment score curve, the horizontal black dotted line indicates the baseline at ES = 0, and the black vertical lines mark the positions of the member genes of this gene set within the ranked list. The bottom part of the figure shows the distribution of rank values for all genes, where genes corresponding to the red section are highly expressed in the high-expression group, and genes corresponding to the blue section are highly expressed in Group B. The signal-to-noise ratio (Signal2noise, the ranking method selected earlier) for each gene is displayed as a colored area plot.

**Figure 11 genes-16-01389-f011:**
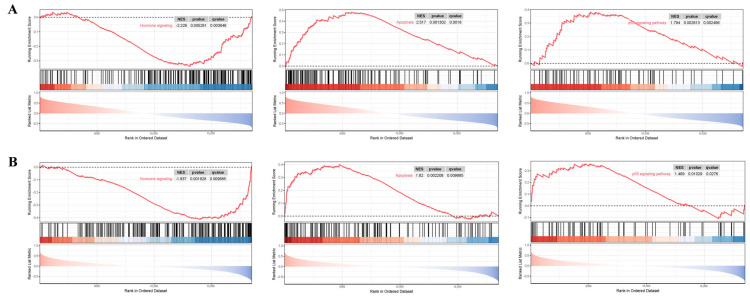
GSEA for the ATP2A2 diagnostic gene. (**A**) GSEA analysis of the ATP2A2 gene in the BBD group; (**B**) GSEA analysis of the ATP2A2 gene in the BC group. The red line in the figure represents the running enrichment score curve, the horizontal black dotted line indicates the baseline at ES = 0, and the black vertical lines mark the positions of the member genes of this gene set within the ranked list. The bottom part of the figure shows the distribution of rank values for all genes, where genes corresponding to the red section are highly expressed in the high-expression group, and genes corresponding to the blue section are highly expressed in Group B. The signal-to-noise ratio (Signal2noise, the ranking method selected earlier) for each gene is displayed as a colored area plot.

**Figure 12 genes-16-01389-f012:**
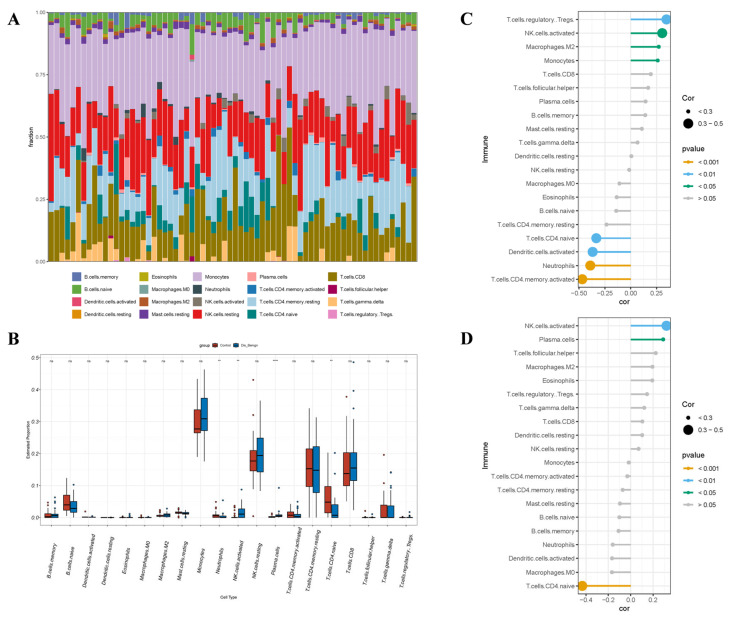
The interrelation between hub genes and immune cell (IC) infiltration within the BBD cohort. (**A**) The infiltrating ICs are depicted in stacked bar charts; (**B**) The violin plot denotes that the BBD cohort exhibits considerably varied categories of Ics, in the box plot, the red points represent outliers from the Control group, while the blue points represent outliers from the Dis_design group; (**C**) The interrelation between *ARRDC1* and ICs in the BBD population; (**D**) The interrelation between *ATP2A2* and ICs in the BBD population. Note: * *p* < 0.05, ** *p* < 0.01, *** *p* < 0.001, **** *p* < 0.0001.

**Figure 13 genes-16-01389-f013:**
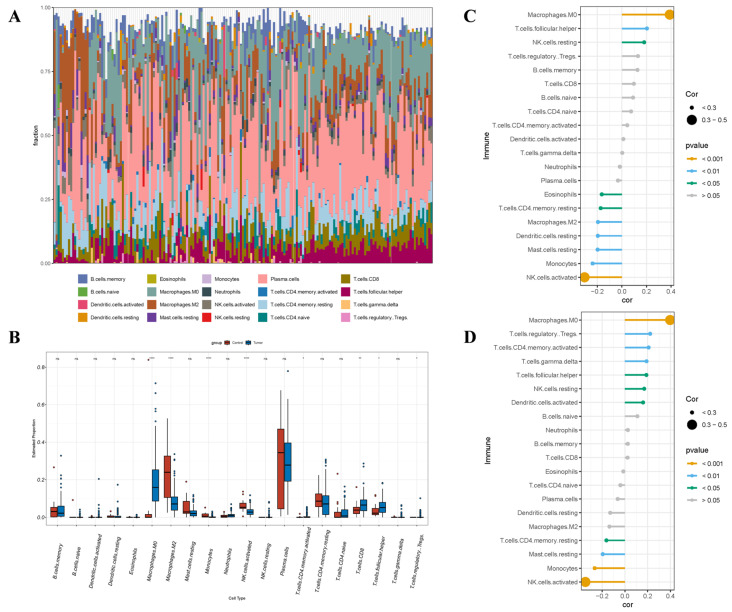
The interrelation between hub genes and immune cell (IC) infiltration within the BC cohort. (**A**) The infiltrating ICs are depicted in stacked bar charts; (**B**) The violin plot denotes that the BC cohort exhibits considerably varied categories of ICs; (**C**) The interrelation between *ARRDC1* and ICs in the BC population; (**D**) The interrelation between *ATP2A2* and ICs in the BC population. Note: * *p* < 0.05, ** *p* < 0.01, *** *p* < 0.001, **** *p* < 0.0001.

**Table 1 genes-16-01389-t001:** Detailed information on the public datasets used in this study.

Dataset	Sample Size		Platform	Note
GSE27562	Healthy control 31	Benign breast abnormalities 37	GPL570	Test dataset
GSE42568	Healthy control 17	Breast cancer 104	GPL570	Test dataset
GSE61304	Healthy control 4	Breast cancer 58	GPL570	Test dataset
TCGA-BRCA	Paracancerous 113	Breast cancer 1106	TCGA	Validation dataset

## Data Availability

The data presented in this study are available on request from the corresponding author due to privacy protection of human subjects.

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
