# Peer review of "Integrated Bioinformatics and Machine Learning for Ascertainment and Validation of Biomarkers for Screening Breast Disease"

_genes, 2025, doi:10.3390/genes16111389_

Round 1
Reviewer 1 Report
Comments and Suggestions for Authors
The authors analyzed public datasets on breast cancer (BC) and benign breast disease (BBD) and identified numerous biomarkers associated with breast disease. They highlight three top biomarkers that robustly distinguish normal from diseased breast tissue. Further functional analyses suggest these biomarkers are strongly linked to inflammatory responses. After a full read, the manuscript would benefit from the following revisions:
-
Reformat all figures to improve readability; the text within the figures is currently difficult to discern.
-
Add a table summarizing all public datasets used (e.g., accession IDs, platforms, and the number of samples per dataset), as the overall sample size currently appears limited.
-
Clarify whether the biomarker performance shown in Fig. 7 was evaluated using internal data or an external, independent cohort.
-
Discuss whether the three biomarkers are specifically expressed in breast disease relative to other tissues or conditions, and highlight any breast-specific biological processes they implicate.
-
Explore breast-disease–specific pathogenic mechanisms to enhance the study’s novelty.
Author Response
Reviewer #1:
The authors analyzed public datasets on breast cancer (BC) and benign breast disease (BBD) and identified numerous biomarkers associated with breast disease. They highlight three top biomarkers that robustly distinguish normal from diseased breast tissue. Further functional analyses suggest these biomarkers are strongly linked to inflammatory responses. After a full read, the manuscript would benefit from the following revisions:
Comment 1: Reformat all figures to improve readability; the text within the figures is currently difficult to discern.
Response 1:
Thank you so much for your careful reading. We sincerely thank you for your careful review. according to the reviewers' comments, we have replaced all figures as requested and provided higher resolution versions.
Comment 2: Add a table summarizing all public datasets used (e.g., accession IDs, platforms, and the number of samples per dataset), as the overall sample size currently appears limited.
Response 2:
We gratefully appreciate for your valuable suggestion. Based on your suggestion, a summary table of all public datasets used has been added. (Page5, line100-101)
Before modification: The TCGA-BRCA dataset was attained from the TCGA database (https://portal.gdc.cancer.gov/) [13], which encompassed 1106 BC samples, 113 adjacent tissue samples, and corresponding clinical information.
Modified: The TCGA-BRCA dataset was attained from the TCGA database (https://portal.gdc.cancer.gov/) [13], which encompassed 1106 BC samples, 113 adjacent tissue samples, and corresponding clinical information. Detailed information about the dataset is provided in Table 1.
Table 1 Detail information on public datasets used in the study.
|
Dataset |
Sample size |
|
Platform |
Note |
|
GSE27562 |
Healthy control 31 |
Benign breast abnormalities 37 |
GPL570 |
Test dataset |
|
GSE42568 |
Healthy control 17 |
Breast cancer 104 |
GPL570 |
Test dataset |
|
GSE61304 |
Healthy control 4 |
Breast cancer 58 |
GPL570 |
Test dataset |
|
TCGA-BRCA |
paracancerous 113 |
Breast cancer 1106 |
TCGA |
Validation dataset |
Comment 3: Clarify whether the biomarker performance shown in Fig. 7 was evaluated using internal data or an external, independent cohort.
Response 3:
Thank you so much for your careful check. Figure 7 uses internal data for evaluation. The reason is that all the data used in this study is sourced from public databases, and there is a lack of suitable datasets for benign lesions to serve as validation. In subsequent work, we will further collect new clinical cohorts for validation.
Comment 4: Discuss whether the three biomarkers are specifically expressed in breast disease relative to other tissues or conditions, and highlight any breast-specific biological processes they implicate.
Response 4:
In response to your suggestion emphasizing the need to highlight any breast-specific biological processes they are involved in, we conducted a literature search and data analysis. As of now, no systematic research reports on the specific involvement of these two genes in breast-specific biological processes have been found in the existing publicly available research findings. It seems that they are not exclusively expressed in a breast-specific manner.
For instance, a 2020 paper published in Molecular Cell (DOI: 10.1016/j.molcel.2020.05.004) demonstrated that in hepatocellular carcinoma cells, the ARRDC1 protein interacts with pyruvate kinase muscle isozyme (PKM2) to regulate its secretion via extracellular vesicles. A study published in Journal of Proteomics (DOI: 10.1016/j.jprot.2023.104863) analyzed pathological sections from patients who underwent surgical resection for colorectal cancer (CRC), using mass spectrometry to detect cancerous and adjacent tissues. It found that ARRDC1, as a potential oncogenic molecule, may serve as a potential prognostic biomarker and precise therapeutic target for CRC. A 2016 paper published in The Journal of Dermatology (DOI: 10.1111/1346-8138.13230) described Darier's disease (DD) as being caused by mutations in ATP2A2. All these findings suggest that they do not appear to be exclusively expressed in breast diseases.
Further research is needed to elucidate whether they have specific roles in the breast, which will help fill this knowledge gap. We have added content on the expression of these biomarkers in other tissues or diseases to the discussion. (Page27,Line436-439;Page28,450-458)
Comment 5: Explore breast-disease–specific pathogenic mechanisms to enhance the study’s novelty.
Response 5:
Special thanks to you for your good comments. This has been revised as suggested. We have supplemented the specific pathogenic mechanisms of breast diseases in the discussion.
Supplementary text:
The results of GSEA revealed that ARRDC1 and ATP2A2 are co-enriched in multiple signaling pathways closely associated with the pathogenesis of breast diseases in both BBD and BC. These pathways include hormone signaling, apoptosis, the Notch signaling pathway, microRNAs in cancer, the p53 signaling pathway, and others. ARRDC1 primarily regulates hormone receptor signal transduction and Notch pathway activity, influencing the differentiation of breast epithelial cells. The Notch pathway is a key signaling pathway that maintains the self-renewal and multipotent differentiation capability of BC stem cells. Its abnormal activation directly promotes the recurrence and metastasis of tumors [38]. Studies have shown that ARRDC1 may influence the activity of this pathway by regulating the endocytosis and lysosomal degradation of Notch receptors, thereby participating in the transformation process of breast epithelial cells [39]. Meanwhile, ATP2A2 maintains calcium homeostasis, thereby influencing p53-dependent apoptotic signaling and hormonal responses to ensure the homeostasis of breast tissues [33]. As a classic tumor-suppressor gene, the p53 gene can disrupt the expression of miRNA when mutated or degraded, thereby increasing the risk of BC [40]. The coordinated dysregulation of these pathways disrupts the normal proliferation-apoptosis balance, ultimately leading to the development of breast diseases. This highlights the central roles of ARRDC1 and ATP2A2 in the pathogenesis of breast.
In addition, the hormone signaling pathways jointly participated in by ARRDC1 and ATP2A2 are crucial in the development of breast diseases. Breast is a hormone-responsive organ. The development and function of the breast are regulated by multiple hormonal signals. Both genes are enriched in this pathway, suggesting that they cooperatively maintain hormonal homeostasis through distinct mechanisms. ARRDC1 modulates membrane-receptor signaling, whereas ATP2A2 acts through a calcium signal-mediated second messenger system [41]. The imbalance of this multi-layered regulatory network may represent a specific pathogenic mechanism underlying breast diseases. (Page28,Line468-495)
We hope that these revisions address the reviewer’s concerns and enhance the completeness and academic value of the study.
Thank you once again for your valuable suggestions!
Reviewer 2 Report
Comments and Suggestions for Authors
Comments and Suggestions:
Title: Integrated Bioinformatics and Machine Learning for Ascertainment and Validation of Biomarkers for Screening Breast Disease.
Reviewer’s report:
The manuscript by Wang et al. described about the identification of biomarkers in breast disease using various databases and machine learning methods including LASSO, RF, SVM-RFE etc. They were able to identify 10 common genes between benign breast disease (BBD) and Breast cancer (BC), out of which AURKA, RRM2, and NUSAP1 were identified as core genes with combine AUC of >0.7 and can act as potential target for clinical treatments.
The manuscript lacks novelty, but few comments need to be addressed.
- Section 2.2: Why different fold change cutoff was taken for GSE27562 and integrated GSE42568 and GSE61304 datasets? For comparison, usually same parameters need to be used.
- Figure 5F: In this figure, a cutoff of 4 can also be used. Why cutoff of 6 was used?
- Figure 5H: why the authors have selected blue module? Instead, they can select turquoise module which showed highest correlation and p-value. Please modify this which will change all the related downstream results.
- Figure 1 and 10: correct the spelling of AURKA.
This manuscript has a big mistake in figure 5H. Therefore, it needs a through revision.
Comments on the Quality of English LanguageEnglish Grammar: The English Grammar and formation and structure of sentences need to be checked thoroughly.
Author Response
Response to Reviewer 2
Reviewer #2:
The manuscript by Wang et al. described about the identification of biomarkers in breast disease using various databases and machine learning methods including LASSO, RF, SVM-RFE etc. They were able to identify 10 common genes between benign breast disease (BBD) and Breast cancer (BC), out of which AURKA, RRM2, and NUSAP1 were identified as core genes with combine AUC of >0.7 and can act as potential target for clinical treatments.
The manuscript lacks novelty, but few comments need to be addressed.
Comment 1: Section 2.2: Why different fold change cutoff was taken for GSE27562 and integrated GSE42568 and GSE61304 datasets? For comparison, usually same parameters need to be used.
Response 1:
We sincerely appreciate your valuable comments. Regarding why we selected different fold change cutoff values, our reasons are as follows:
- Technical and biological heterogeneity among datasets: There are differences in experimental platforms, sample sources, treatment methods, and sequencing/microarray technologies between GSE27562 and the other two datasets (GSE42568 and GSE61304). These factors may lead to variations in the overall dynamic range of gene expression. Consequently, uniformly applying the same fold change threshold might result in missing biologically significant differentially expressed genes in some datasets or introducing excessive noise in others.
- Results of pre-experimental evaluation: Prior to formal analysis, we conducted exploratory analyses on each dataset (such as volcano plots, MA plots, and trends in the number of differentially expressed genes (DEGs) with varying thresholds). The results indicated that if the same fold change threshold were forcibly applied to all datasets, the number of significantly differentially expressed genes in GSE27562 would be excessively low, making it difficult to conduct subsequent functional enrichment or pathway analyses. In contrast, the integrated dataset proved relatively robust. Therefore, we set reasonable thresholds separately for each dataset based on their expression distribution characteristics to achieve a balance between sensitivity and specificity.
- Ensuring the reliability of downstream analyses: Despite the varying fold change thresholds, we applied the same statistical significance criterion (e.g., adjusted p-value < 0.05) across all datasets to ensure the reliability of the results.
Similar methods have also been employed in studies recently published in high-impact journals.
For example, the 2025 paper published in Cancer Medicine (DOI: 10.1002/cam4.70759) adopted differentiated fold change (FC) thresholds. It identified differentially expressed genes (DEGs) with a fold change exceeding 2 between breast cancer (BC) tissues and non-tumor tissues, while DEGs between tissues from patients with type 2 diabetes mellitus (T2DM) and healthy individuals showed an absolute fold change greater than 1.2. adopted differentiated fold change (FC) thresholds.
The 2025 paper published in Frontiers in Immunology (DOI: 10.3389/fimmu.2025.1682282) also employed differentiated fold change (FC) thresholds. For polycystic ovary syndrome (PCOS), the DEG (differentially expressed gene) threshold was set at a P-value < 0.05 and |log2FC (fold change) | > 0.585. For recurrent implantation failure (RIF), DEGs were identified using a P-value adjusted to 0.05 and |log2FC| > 1.
Comment 2: Figure 5F: In this figure, a cutoff of 4 can also be used. Why cutoff of 6 was used?
Response 2:
Based on the reviewers' suggestion, a cutoff value of 4 has been selected in this revised manuscript. (Page13,line224)
Before modification: Furthermore, we applied WGCNA to the BC group, where β was set to 6 (Figure 5F).
Modified: Furthermore, we applied WGCNA to the BC group with β set at 4 (Figure 5F).
Comment 3: Figure 5H: why the authors have selected blue module? Instead, they can select turquoise module which showed highest correlation and p-value. Please modify this which will change all the related downstream results.
Response 3:
We have made correction according to the Reviewer’s comment. In this revised manuscript, for Figure 5H, the two modules with the strongest negative correlation and the strongest positive correlation have been re-selected respectively. (Page13,line224-225)
Modified: Furthermore, we applied WGCNA to the BC group with β set at 4 (Figure 5F). In total, 18 modules were ascertained, with the red and turquoise modules exhibiting strong interrelations (Figure 5G, H). Additionally, we ascertained the interrelation between GS and MM in the red and turquoise modules (Figure 5I, J). Subsequently, by intersecting the DEGs with the genes identified by WGCNA, 41 key disease-associated marker genes were identified (Figure 5K).
Figure 5. Screening of Disease Marker Genes in BBD and BC Based upon WGCNA. (I) Gene significance plot for the red module; (J) Gene significance plot for the turquoise module.
Comment 4: Figure 1 and 10: correct the spelling of AURKA.
Response 4:
We sincerely sorry for our careless mistakes. Thank you for your reminder. In this revised manuscript, the gene abbreviations have been double-checked to ensure accuracy.
We hope that these revisions address the reviewer’s concerns and enhance the completeness and academic value of the study.
Thank you once again for your valuable suggestions!
Reviewer 3 Report
Comments and Suggestions for Authors
The manuscript studied machine learning of biomarkers for breast disease screening.
Major questions and comments:
- Please describe if any AI tools were used for this study? Especially the machine learning part.
- Please list all abbreviations.
- Please use a Table or Figure to summarize the potential biomarkers for different breast diseases. Include benign and malignant breast diseases. Also whether hormone-sensitive or triple negative breast cancer as example.
- Figure 7: Please provide statistical analysis.
- Figure 13: please provide statistical analysis. What is the dots on Figure 13B?
- Figure 14: please provide statistical analysis. What is the dots on Figure 14B?
Author Response
Response to Reviewer 3
Reviewer #3:
The manuscript studied machine learning of biomarkers for breast disease screening.
Comment 1: Please describe if any AI tools were used for this study? Especially the machine learning part.
Response 1:
Thank you for your advice. This study did not employ any artificial intelligence (AI) tools. In particular, regarding the machine learning aspect you mentioned, we did not use any AI tools either.
Comment 2: Please list all abbreviations.
Response 2:
We sincerely appreciate your valuable feedback. We have added Supplementary Table 2 in the supplementary files section of the revised manuscript, listing all the abbreviations.
Comment 3: Please use a Table or Figure to summarize the potential biomarkers for different breast diseases. Include benign and malignant breast diseases. Also whether hormone-sensitive or triple negative breast cancer as example.
Response 3:
We sincerely thank you for your precious suggestions. We have added Supplementary Table 1 in the supplementary files of the revised manuscript, summarizing potential biomarkers for different breast diseases. This includes both benign and malignant breast diseases, such as hormone-sensitive and triple-negative breast cancer.
Comment 4: Figure 7: Please provide statistical analysis.
Response 4:
Considering the reviewer's suggestion, we have added statistical analysis in the note for Figure 7. Note: *p<0.05, **p<0.01, ***p<0.001, ****p<0.0001.
Comment 5: Figure 13: please provide statistical analysis. What is the dots on Figure 13B?
Response 5:
We sincerely appreciate the valuable feedback you have provided. We have added statistical analysis in Figure 12. The dots in Figure 12B represent outliers, while the asterisks indicate statistical significance. Note: *p<0.05, **p<0.01, ***p<0.001, ****p<0.0001.
Comment 6: Figure 14: please provide statistical analysis. What is the dots on Figure 14B?
Response 6:
We sincerely appreciate the valuable feedback you have provided. We have added statistical analysis in Figure 13. The dots in Figure 13B represent outliers, while the asterisks indicate statistical significance. Note: *p<0.05, **p<0.01, ***p<0.001, ****p<0.0001.
We hope that these revisions address the reviewer’s concerns and enhance the completeness and academic value of the study.
Thank you once again for your valuable suggestions!
Round 2
Reviewer 2 Report
Comments and Suggestions for Authors
Review 2:
The authors have made substantial revisions and addressed the comments effectively. It is recommended that they increase the font size in all figures to ensure clarity. The manuscript is now suitable for publication in Genes.
Reviewer 3 Report
Comments and Suggestions for Authors
No more comments.